# A conserved protein tyrosine phosphatase, PTPN-22, functions in diverse developmental processes in *C. elegans*

**Shaonil Binti**[ID], **Adison G. Linder, Philip T. Edeen**[ID], **David S. Fay**[ID]*

Department of Molecular Biology, College of Agriculture, Life Sciences and Natural Resources, University of Wyoming, Laramie, Wyoming, United States of America

* davidfay@uwyo.edu

## Abstract

Protein tyrosine phosphatases non-receptor type (PTPNs) have been studied extensively in the context of the adaptive immune system; however, their roles beyond immunoregulation are less well explored. Here we identify novel functions for the conserved *C. elegans* phosphatase PTPN-22, establishing its role in nematode molting, cell adhesion, and cytoskeletal regulation. Through a non-biased genetic screen, we found that loss of PTPN-22 phosphatase activity suppressed molting defects caused by loss-of-function mutations in the conserved NIMA-related kinases NEKL-2 (human NEK8/NEK9) and NEKL-3 (human NEK6/NEK7), which act at the interface of membrane trafficking and actin regulation. To better understand the functions of PTPN-22, we carried out proximity labeling studies to identify candidate interactors of PTPN-22 during development. Through this approach we identified the CDC42 guanine-nucleotide exchange factor DNBP-1 (human DNMBP) as an in vivo partner of PTPN-22. Consistent with this interaction, loss of DNBP-1 also suppressed *nekl*-associated molting defects. Genetic analysis, co-localization studies, and proximity labeling revealed roles for PTPN-22 in several epidermal adhesion complexes, including *C. elegans* hemidesmosomes, suggesting that PTPN-22 plays a broad role in maintaining the structural integrity of tissues. Localization and proximity labeling also implicated PTPN-22 in functions connected to nucleocytoplasmic transport and mRNA regulation, particularly within the germline, as nearly one-third of proteins identified by PTPN-22 proximity labeling are known P granule components. Collectively, these studies highlight the utility of combined genetic and proteomic approaches for identifying novel gene functions.

## Author summary

The human protein, PTPN22, has been studied extensively within the context of the adaptive immune system and mutations that affect PTPN22 function are implicated in multiple autoimmune disorders. Curiously, PTPN22 is conserved throughout the animal kingdom, including in the invertebrate roundworm *Caenorhabditis elegans*, which lack an adaptive immune system. Using several non-biased approaches to uncover novel protein

**Data Availability Statement:** The direct link for all (raw) proteomic data is: https://massive.ucsd.edu/ProteoSAFe/dataset.jsp?task=121d79147e584f629e892d70e073112f All other

data are already contained in the supplementary files and information.

**Funding:** This project was supported by NIH R35 GM136236 to DSF, SB, PTD and AGL (University of Wyoming) and by an Institutional Development Award (IDeA) from the National Institute of General Medical Sciences of the National Institutes of Health (P20GM103432) to DSF, SB, PTD and AGL (University of Wyoming). The funders had no role in study design, data collection and analysis, decision to publish, or preparation of the manuscript.

**Competing interests:** The authors have declared that no competing interests exist.

functions, we identified roles for the *Caenorhabditis elegans* PTPN22 ortholog, PTPN-22, in a several different cell biological processes including functions related to epithelial cell and tissue connectivity. These studies, along with our analysis of the cellular and subcellular locations in which PTPN-22 is present during development, indicate that the worm PTPN-22 carries out a diverse range of functions. Our studies indicate that, in addition to critical roles within the immune-system, PTPN22 family members may carry additional functions that are conserved throughout the animal kingdom, a finding supported by several studies carried out on human PTPN22 suggesting roles in cell-cell adhesion.

## Introduction

Reversible tyrosine phosphorylation is a key mechanism for regulating diverse cellular processes including differentiation, proliferation, apoptosis, metabolism, and signal transduction. The steady-state level of tyrosine-phosphorylated proteins is regulated by the coordinated effects of protein tyrosine kinases (PTKs) and protein tyrosine phosphatases (PTPs). Of the 518 protein kinases in humans, 90 (17%) are classified as PTKs, whereas of the 200 protein phosphatases, 108 (54%) are classified as PTPs [1–3]. Despite their greater abundance, PTPs have garnered less attention from researchers than PTKs. Nevertheless, it is known that dysfunction of PTPs can disrupt the homeostasis of tyrosine-phosphorylated proteins, contributing to a spectrum of disorders including cancers and neuronal and autoimmune diseases [4–9]. Notably, among the 237 protein phosphatases in *C. elegans*, 94 are presumed to be PTPs and over 50% of *C. elegans* phosphatases have human orthologs [10,11].

Protein tyrosine phosphatase non-receptor type (PTPN) is a subset of the PTP family comprising 17 protein tyrosine phosphatases in humans. One well-studied member of this family, PTPN22, has been investigated primarily in cells of hematopoietic origin including lymphocytes, monocytes, natural killer cells, and platelets [12–17]. Correspondingly, one of the most well-characterized functions of PTPN22 is the suppression of T-cell activation. In conjunction with C-terminal Src tyrosine kinase (CSK), PTPN22 is responsible for dephosphorylating T-cell signaling receptors including LCK, FYN, CD3ζ, and ZAP-70 [18,19]. Moreover, a missense polymorphism in PTPN22, which leads to a R620W substitution in the C terminus of PTPN22, is a common risk factor for multiple autoimmune diseases including type I diabetes mellitus, systemic lupus erythematosus, and rheumatoid arthritis [20–24]. PTPN22 expression is also detected in epithelial, endothelial, muscle, and nervous tissue, suggesting additional roles for this phosphatase, although it has been less well studied in these contexts [14–16,25,26].

PTPs also contribute to the formation and maintenance of adhesion complexes in human cells. Several members of the PTPN family including PTPN11, PTP-PEST (PTPN12), and PTPN22 are recognized as key regulators of integrin function. Integrins are transmembrane αβ heterodimeric receptors that connect the extracellular matrix to the actin cytoskeleton, and their activation is regulated by phosphorylation to control cell movement and adhesion [27–30]. In human T cells, PTPN22 inhibits signaling by integrin subunit alpha L (ITGAL), which is also known as integrin lymphocyte function-associated antigen 1 (LFA-1), and loss of PTPN22 function results in increased ITGAL–mediated cell adhesion [31]. In PTPN22$^{-/-}$ mice, regulatory T cells also exhibit an increase in ITGAL–dependent adhesion, and these mice show an increase in platelet-specific $\alpha_{IIb}\beta_3$ integrin activation along with increased platelet spreading and aggregation [32,33]. Together these findings implicate PTPN22 in cell

adhesion, although analogous functions for PTPN22 beyond the immune system remain largely unexplored.

As described below, we identified the *C. elegans* ortholog of PTPN22 in a genetic screen for genes that affect the *C. elegans* molting process. Molting is an essential developmental process in nematodes and other members of the ecdysozoan group, allowing for organismal growth and adaptation to new environments [34]. Molting occurs at the termination of each of the four larval stages (L1–L4), wherein a new cuticle—an apical extracellular matrix surrounding the epidermis—is synthesized and the old cuticle is released. Many different types of proteins have roles in the molting process including signal transducers, transcriptional regulators, structural components and modifiers of the cuticle, cell–ECM adhesion complexes, and proteins that affect membrane trafficking [34–36]. Previous work in our laboratory identified two conserved Serine–Threonine-specific NIMA-related kinases, NEKL-2 (human NEK8/NEK9) and NEKL-3 (human NEK6/NEK7) (collectively referred to as the NEKLs)—along with their three ankyrin repeat binding partners, MLT-2 (human ANKS6), MLT-3 (human ANKS3), and MLT-4 (human INVS) (collectively referred to as the MLTs)—as being required for molting [37,38].

Ongoing research continues to uncover the precise mechanisms by which NEKLs and MLTs affect the molting process in *C. elegans*. Most notably, we have shown that NEKLs and their human counterparts regulate several aspects of membrane trafficking [39,40]. Further insights into the functions of NEKLs came from the isolation of genetic suppressors of *nekl* molting defects, which include core components of the endocytic machinery and several closely associated factors that regulate actin filamentation, including the Rho-family GTPase CDC-42 and its effector SID-3, which is the ortholog of human TNK2 (tyrosine kinase non receptor 2; also known as ACK1, activated CDC42-associated kinase 1) [40–42]. In this study, we identified mutations affecting PTPN-22, which we showed binds to the Rho-guanine nucleotide exchange factor (GEF) DNBP-1, loss of which also suppressed *nekl*-associated molting defects. Using genetic, proteomic, and cell biological approaches, we have implicated PTPN-22 in a range of functions including effects on epidermal adhesion complexes, cytoskeletal proteins, and potential germline expression functions, thereby expanding substantially on the known functions of PTPN22 family members.

## Results

### Loss of PTPN-22 suppresses molting defects in *nekl* mutants

We previously showed that *nekl-2* and *nekl-3* are required for molting in *C. elegans*, as strong loss-of-function mutations in either gene cause molting defects in early larval development [37,38]. In contrast, the weak loss-of-function mutations *nekl-2(fd81)* and *nekl-3(gk894345)* do not exhibit phenotypes individually, but when combined they lead to ~98% of double-mutant larvae arresting at the L2/L3 molt (Fig 1A and 1B) [37]. *nekl-2(fd81)*; *nekl-3(gk894345)* homozygotes (hereafter referred to as *nekl-2*; *nekl-3*) can be maintained in the presence of an extra-chromosomal array (*fdEx286*) that contains wild-type copies of *nekl-3* along with a broadly expressed GFP reporter (SUR-5::GFP; Fig 1A) [37].

To identify proteins that functionally interact with NEKL kinases, we carried out a forward genetic screen to identify suppressors of molting defects in *nekl-2*; *nekl-3* mutants [41]. From this screen we identified allele *fd269*, which led to ~50% of *nekl-2*; *fd269*; *nekl-3* mutants reaching adulthood (Fig 1B). Using whole-genome sequencing together with the Sibling Subtraction Method [41], we identified an insertion in the sixth exon of *ptpn-22* (Y41D4A.5) that led to a frameshift after D238 of PTPN-22 followed by subsequent stop codons (Fig 1D and S1 File). To determine if the alteration in *ptpn-22* led to *nekl-2*; *nekl-3* suppression, we used CRISPR

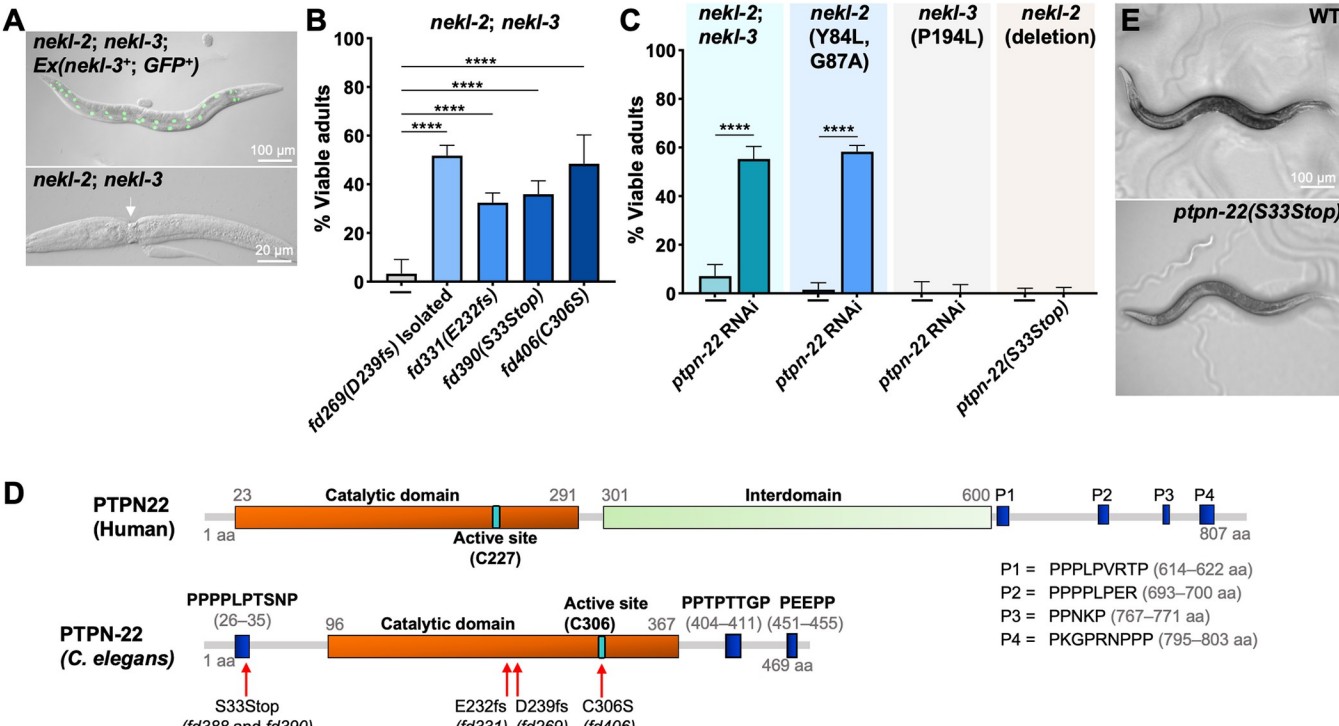

**Fig 1. Loss of *ptpn-22* can suppress *nekl*-associated molting defects.** (A) Merged fluorescence and DIC images of *nekl-2(fd91)*; *nekl-3(gk894345)* worms in the presence (top) and absence (bottom) of the extrachromosomal array (*fdEx286*), which contains wild-type *nekl-3* and SUR-5::GFP. Note the molting defective *nekl-2; nekl-3* double mutant in the lower panel, which exhibits a mid-body constriction due to a failure to shed its old cuticle. (B,C) Bar graphs indicating the percentage of worms that developed into viable adults for the indicated genotypes including *ptpn-22* genetic mutations and *ptpn-22(RNAi)*. (D) Diagram highlighting the structural features of human PTPN22 and *C. elegans* PTPN-22 proteins including the catalytic domains (orange), active sites (turquoise), interdomain (light green; PTPN22 only), and proline-rich regions (PR.1–3; blue). Amino acid sequences of the proline-rich domains are also provided. Also indicated are the locations and effects of *ptpn-22* alleles shown in Fig 1B. (E) Bright-field images of a wild-type worm and mutant worm carrying the *ptpn-2(S33Stop)* mutation. Error bars represent 95% confidence intervals. Fisher's exact test was used to calculate p-values; ****p < 0.0001. Raw data are available in the S7 File. The sequences for *ptpn-22* alleles can be found in the S1 File. aa, amino acid.

methods to generate a frameshift mutation after L231 (*fd331*) (Fig 1D and S1 File). Likewise, CRISPR targeting the N terminus of PTPN-22 yielded a 27-bp insertion in the second exon of PTPN-22, leading to premature stop codons after position T32 (*fd390*) (Fig 1D and S1 File). Notably, both mutations induced suppression of *nekl-2; nekl-3* mutants, albeit at slightly lower levels than *fd269* (Fig 1B).

Additionally, RNA interference (RNAi) of *ptpn-22* using dsRNA injection methods led to ~50% suppression of *nekl-2; nekl-3* lethality, further indicating that loss of PTPN-22 function was responsible for the suppression of *nekl-2; nekl-3* molting defects (Fig 1C). However, whereas *ptpn-22(RNAi)* effectively suppressed the moderate loss-of-function allele of *nekl-2*, *fd91* (Y84L, G87A), it failed to suppress molting defects associated with a moderate loss-of-function allele of *nekl-3*, *sv3* (P194L) (Fig 1C). Furthermore, another CRISPR-generated mutation that led to a stop codon after T32 (*fd388*) failed to suppress molting defects in a null allele of *nekl-2* (*gk839*) (Fig 1C). Collectively, our data indicate that loss of *ptpn-22* can suppress weak and some moderate loss-of-function *nekl* mutations but not strong loss-of-function mutations, a profile exhibited by several other previously described *nekl* suppressors [40,43,44].

Human PTPN22 is an 807-amino-acid (aa) protein containing a tyrosine phosphatase catalytic domain at the N terminus followed by an interdomain and several C-terminal proline-

rich motifs (P1–P4) (Fig 1D) [45]. As compared to its human ortholog, *C. elegans* PTPN-22 is relatively small (469 aa) and contains a single proline-rich region near its N terminus, followed by the catalytic domain and two additional proline-rich regions in its C terminus (Fig 1D). Sequence alignment between PTPN-22 and orthologs in other organisms indicates conservation primarily within the catalytic domain (S1 Fig); PTPN-22 is 38% identical to human PTPN22 and 59% similar in this region. Animals carrying the *ptpn-22(S33Stop)* mutation in an otherwise wild-type (N2) background did not exhibit obvious morphological or developmental defects (Fig 1E), indicating that PTPN-22 is not essential under normal growth conditions. To determine if loss of PTPN-22 catalytic activity is critical for its suppression of *nekl* molting defects, we mutated the active site cysteine residue (C306), equivalent to C227 in human PTPN22, using CRISPR/Cas9 methods [19,46]. We observed that *ptpn-22(C306S)* suppressed *nekl-2*; *nekl-3* molting defects by ~50% (Fig 1B), indicating that it is the loss of PTPN-22 phosphatase activity that leads to the suppression of *nekl-2*; *nekl-3* molting defects.

## Proximity labeling identifies candidate in vivo PTPN-22 interactors

Although mammalian members of the PTPN22 family have been studied extensively in the context of the adaptive immune system, those findings did not suggest an obvious mechanism by which *C. elegans* PTPN-22 might functionally interact with NEKL proteins in the context of molting. We therefore took a non-biased approach to identify potential partners and targets of *C. elegans* PTPN-22 by mapping its in vivo (proximate) interactome using biotin-based proximity labeling methods. Proximity labeling (also referred to as BioID) methods can be effective for detecting high-affinity binding partners as well as weak or transient interactors, including the substrates of protein kinases and phosphatases [47–55]. Additionally, proximity labeling can provide insights into tissue expression, subcellular localization, and cellular functions.

In proximity labeling, a protein of interest is fused to an *Escherichia coli*–derived biotin ligase, BirA, and the resulting fusion protein is expressed in cells using either a native or heterologous promoter. In the presence of ATP, BirA converts non-reactive biotin into biotinoyl-5′-adenylate, which reacts with lysine residues on nearby (in proximity) proteins to form covalent linkages to biotin (Fig 2A) [56,57]. For our analysis, we used the non-specific biotin ligase TurboID, an improved version of the BirA enzyme with rapid labeling kinetics, which was fused to the C terminus of PTPN-22 (referred to hereafter as PTPN-22::TurboID) using CRISPR-based genome editing [58]. As our control, we used N2 worms, which lack TurboID but contain an endogenous specific biotinylation activity that primarily targets four known carboxylases (MCCC-1, PCCA-1, PYC-1, and POD-2) [53,58–61]. Furthermore, when PTPN-22::TurboID was introduced into *nekl-2; nekl-3* mutants we failed to observe suppression of molting defects (1.0% viability; n = 712), indicating that the fusion retained wild-type levels of function based on the suppression assay.

Four replicates were carried out for both experimental (PTPN-22::TurboID) and control (N2) strains using mixed-stage worm populations. Crude lysates containing biotinylated proteins were subjected to pull-down using streptavidin-conjugated beads followed by washing (Fig 2A). As anticipated, both crude lysates and the corresponding streptavidin-purified samples from N2 and PTPN-22::TurboID exhibited several prominent bands after western blotting that corresponded to the endogenously biotinylated carboxylases (Fig 2B). Encouragingly, PTPN-22::TurboID samples also contained numerous additional biotin-positive bands in both input and pull-down fractions, confirming the functionality of the fused TurboID enzyme (Fig 2B). Purified samples were subjected to on-bead trypsin digestion followed by liquid chromatography–tandem mass spectroscopy (LC-MS/MS) analysis using an established data-independent acquisition pipeline.

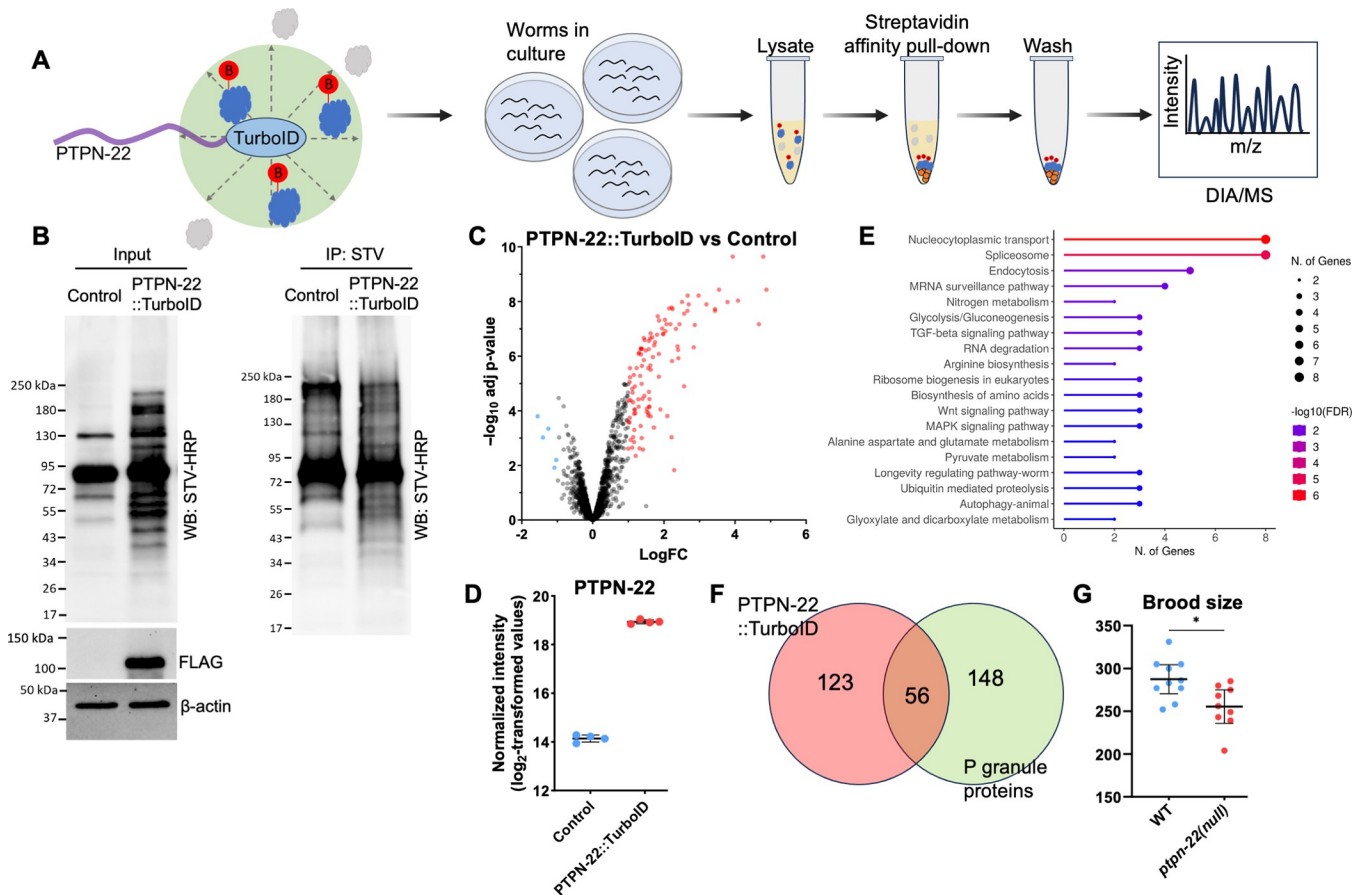

**Fig 2. TurboID-based proximity labeling identifies the PTPN-22 interactome.** (A) Schematic illustrating the proximity labeling study. The C terminus of PTPN-22 was fused to TurboID::3×-FLAG, leading to the biotinylation of proximal proteins. Proximal proteins are depicted in blue, with the resulting biotin modification highlighted in red, whereas proteins located outside the TurboID labeling radius (~10 nm) are represented in gray. PTPN-22::TurboID or N2 control animals were cultured on plates, and subsequent protein extraction was carried out. Biotinylated proteins were pulled down using streptavidin-coated magnetic beads (orange), whereas non-biotinylated proteins were removed through washing steps. Enriched biotinylated proteins were subjected to on-bead digestion, followed by Data-Independent Acquisition (DIA) LC-MS/MS analysis. (B) Western blot (WB; left) shows the input fractions of representative N2 and PTPN-22::TurboID samples probed with streptavidin-HRP. Note additional bands in the PTPN-22::TurboID lysate versus the N2 control. The expression of PTPN-22::TurboID was visualized through an anti-FLAG western blot; antibodies against β-actin were used as a loading control. The pull-down fraction (IP, right) shows N2 and PTPN-22::TurboID samples probed with streptavidin-HRP after enriching for biotinylated proteins using streptavidin-coated beads. (C) Volcano plot highlighting proteins enriched (>2-fold and p-value <0.05) in PTPN-22::TurboID samples (red) versus N2 (blue). (D) Dot plots show the enrichment of PTPN-22 in PTPN-22::TurboID samples; error bars represent standard deviation. (E) KEGG pathway enrichment analysis was performed using ShinyGO 0.80, and the top 19 biological pathways based on fold enrichment are shown [124]. (F) Venn diagram shows the overlap of enriched proteins between PTPN-22::TurboID samples and P granule proteins (S2 File) (G) The dot plot shows the brood size of individual worms in the indicated backgrounds.

We found 112 proteins that were enriched ≥2-fold in the PTPN-22::TurboID samples relative to N2, that were detected in all four PTPN-22::TurboID replicates, and that had adjusted p-values of <0.05 (Fig 2C and S2 File; Sheet, "Fold Change >2.0"). Conversely, only five proteins were enriched in N2 versus PTPN-22::TurboID samples. Moreover, PTPN-22 was the second most highly enriched protein in the PTPN-22::TurboID samples (~28-fold) (Fig 2D). In addition, 67 proteins were detected in at least three of the four PTPN-22::TurboID replicates but were undetected in all four N2 control samples (S2 File; Sheet, "ND Control"), leading to a total of 179 proteins that were designated as enriched in PTPN-22:TurboID samples versus N2 (S2 File; Sheet, "Combined Shortlist"). Estimates suggest that the four biotinylated carboxylases collectively accounted for ~22% of the proteins identified by LC-MS/MS in the N2 samples and ~13% in the PTPN-22::TurboID samples, consistent with the enrichment of

biotinylated proteins by the purification process and with the abundance of biotinylated carboxylases in our western blots relative to PTPN-22::TurboID–specific bands (Fig 2B). These findings further suggest that the presence of the endogenously biotinylated carboxylases did not substantially affect the sensitivity of our approach with respect to detecting PTPN-22::TurboID targets.

False positives in proximity labeling can occur because of non-specific binding by non-biotinylated proteins to beads and because of non-specific (promiscuous) biotinylation by the TurboID-tagged proteins themselves. Examples of the latter may occur in cases where non-specific protein targets are highly abundant, localized within the same compartment(s) as the protein of interest, and because they are susceptible to biotinylation, such as proteins with numerous surface-exposed lysine residues [62,63]. To address this issue, we cross-referenced our shortlist of 179 PTPN-22::TurboID hits with data from a previous study by Artan et al. [53] in which a non-localized GFP::TurboID was expressed at high levels in four different tissues of the worm (intestine, hypodermis, neurons, and muscle). More specifically, we identified 435 proteins that were among the top 200 most abundant proteins in at least one of the four tissues and then looked for overlap with our PTPN-22::TurboID dataset (S2 File, Sheet, "Artan Top 200 four tissues"). Among the 112 proteins showing >2-fold enrichment in the PTPN-22:TurboID study, 62 were also present in the highly enriched dataset; of the 67 proteins in the "ND Control" dataset, 14 were present in the highly enriched dataset (S2 File, "Artan Overlap"). Overall, our analysis suggests that although some of the proteins identified by PTPN-22::TurboID likely represent non-specific targets, many others may correspond to authentic PTPN-22 proximate proteins.

Gene ontology analysis of the 179 PTPN-22::TurboID–associated proteins indicated statistically significant enrichment of proteins acting within various molecular and cellular structures including the actin cytoskeleton, germline P granules, the NatC/N-terminal protein acetyltransferase complex, and the spliceosomal small nuclear ribonucleoprotein (snRNP) complex, among others (S3 File). Enriched biological processes implicated pathways associated with germline functions, protein transport, mRNA processing and regulation, and several signaling pathways (Fig 2E and S3 File). Strikingly, of the 179 proximate interactors identified by PTPN-22::TurboID, 56 (31%) are reported to be components of germline P granules (Fig 2F and S2 File; Sheet, "Overlap with P granules"), RNP condensates that serve as critical regulators of germline gene expression [64]. Notably, we observed expression of PTPN-22::EGFP in the perinuclear region of germline cells in a punctate pattern similar to that reported for P granule proteins (S2 Fig). Consistent with a potential role in the germline, *ptpn-22* mutants had diminished brood sizes relative to wild type (Fig 2G). Collectively, our proximity labeling findings implicate PTPN-22 in a diverse range of molecular and tissue-specific functions. This is consistent with the observed expression of PTPN-22::EGFP and PTPN-22::mScarlet in multiple tissues (S2 Fig and see below) and with RNA expression data available on WormBase, indicating that PTPN-22 is expressed in multiple tissues throughout development [65].

Notably, whereas PTPN-22 was broadly expressed, the NEKLs and MLTs are expressed and required specifically for molting in the hyp7 epidermal syncytium [37–39]. To identify PTPN-22–associated proteins that act within hyp7, we also carried out proximity labeling studies, using three technical replicates, in which PTPN-22::TurboID was expressed under the control of a strong hyp7-specific promoter (Y37A1B.5; $P_{hyp7}$::PTPN-22::TurboID) via a multicopy array (S3A Fig). We validated the functionality of this transgene through western blotting and, as expected, observed increased levels of biotinylated proteins versus N2 control worms (S3A Fig). Using the above LC-MS/MS pipeline, we identified 246 proteins that were enriched ≥2-fold in $P_{hyp7}$::PTPN-22::TurboID versus N2 samples, were present in all three $P_{hyp7}$::PTPN-22::TurboID replicates, and had adjusted p-values of <0.05; three additional proteins

were present in all three P$_{hyp7}$::PTPN-22::TurboID samples but were undetected in all three N2 controls (S3B Fig and S4 File). PTPN-22 was the seventh most enriched protein in the dataset (~24-fold; S3C Fig) and, as expected, P$_{hyp7}$::PTPN-22::TurboID hits exhibited minimal overlap with germline-associated proteins such as P granule components (S3D Fig and S4 File). Similar to our findings for PTPN-22::TurboID, 62/249 proteins enriched in the P$_{hyp7}$::PTPN-22::TurboID dataset overlapped with the list of 435 highly abundant biotinylated proteins (S4 File).

Unlike the PTPN-22::TurboID dataset, however, the volcano plot for P$_{hyp7}$::PTPN-22::TurboID indicated substantially reduced specificity, given that 193 proteins were upregulated ≥2-fold (p <0.05) in N2 versus P$_{hyp7}$::PTPN-22::TurboID samples (compare Fig 2C with S3B Fig). Consistent with this, the four endogenous carboxylases accounted for only ~3% of total identified proteins in the P$_{hyp7}$::PTPN-22::TurboID samples and ~13% in the corresponding N2 controls, suggesting a somewhat higher incidence of non-specific binding by non-biotinylated proteins to the streptavidin beads. Other differences between the two PTPN-22 proximity labeling experiments may be due to differences in expression levels and tissues and to additional differences in the protocols used to enrich for the biotinylated proteins (see Materials and Methods).

An analysis of enriched molecular/cellular GO terms in the P$_{hyp7}$::PTPN-22::TurboID dataset included proteins linked to the actin cytoskeleton, cytoskeletal fibers, and several trafficking compartments (S5 File). Although only 29 proteins overlapped between the PTPN-22::TurboID and P$_{hyp7}$::PTPN-22::TurboID shortlists (S3E Fig), these included the actin-capping proteins CAP-1 and CAP-2, the latter of which genetically interacts strongly with *ptpn-22* (S4 File; Sheet, "Overlap"; and see below). Collectively, our proximity labeling studies provide a foundation for uncovering previously unknown molecular and cellular functions and partners of the PTPN-22 family of proteins within the context of an intact developing organism.

## DNBP-1 associates with PTPN-22 and is a suppressor of *nekl* molting defects

Given that we identified *ptpn-22* as a suppressor of *nekl* molting defects, we were interested in identifying proteins that could functionally connect PTPN-22 to the NEKL–MLT pathway. Interestingly, cross-referencing our list of candidate PTPN-22 interactors with protein–protein interaction data for PTPN-22 available on WormBase [65] yielded a single hit, DNBP-1 (dynamin binding protein 1), which was enriched ~3.3-fold in all four PTPN-22::TurboID samples (adjusted p < 0.0001) (Fig 3A). A prior yeast two-hybrid screen of worm Src homology-3 (SH3) domains identified a high-confidence interaction between PTPN-22 and the third SH3 domain of DNBP-1 [66]. In addition, that screen identified interactions between the first and second SH3 domains of DNBP-1 with DYN-1 (dynamin), an interaction that is conserved in humans [66–68]. We also note that DNBP-1 was slightly enriched (~1.5-fold) in all three P$_{hyp7}$::PTPN-22::TurboID samples, although this result was not statistically significant (S4 File; Sheet, "Raw Data").

DNBP-1 is an ortholog of human DNMBP/Tuba, which functions as guanine exchange factor (GEF; i.e., activator) for the Rho-family GTPase CDC-42 [67,68]. Given that we previously showed that loss of function in CDC-42 and its effector SID-3/TNK2 can suppress *nekl-2*; *nekl-3* molting defects and that CDC-42 becomes hyperactivated in *nekl* mutants, we hypothesized that the loss of a CDC-42 activator might similarly alleviate molting defects in *nekl* mutants [42]. To test this, we generated two CRISPR-based loss-of-function alleles of *dnbp-1*. *dnbp-1(fd385)* introduces a 66-bp insertion into the sixteenth exon of *dnbp-1*, leading to 13 new amino acids after K913, followed by multiple stop codons (S1 File). *dnbp-1(fd386)*

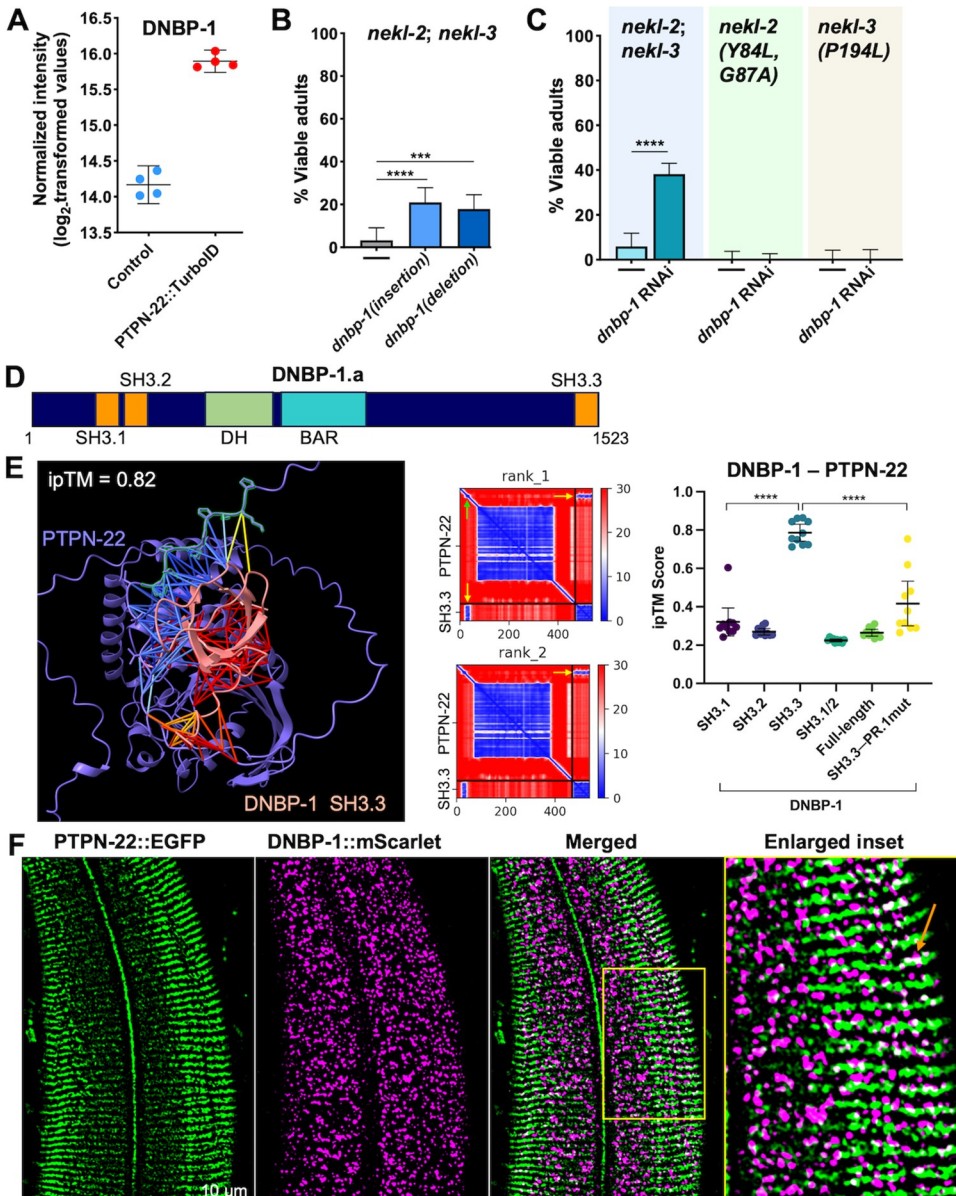

**Fig 3. DNBP-1 associates with PTPN-22, and its loss suppresses *nekl* molting defects.** (A) Dot plot showing DNBP-1 enrichment in all four PTPN-22::TurboID samples; error bars represent standard deviation. (B, C) Bar graphs show the percentage of worms that developed without molting defects in different *nekl* mutants achieved by reducing DNBP-1 activity through either loss-of-function mutations (B) or RNAi (C). Fisher's exact test was used to calculate p-values; ****p < 0.0001 and ***p < 0.001. (D) Schematic of DNBP-1A isoform showing structural domains. (E) One of the best predicted models by AlphaFold-multimer showing the predicted binding interaction between PTPN-22 (in blue) and the SH3.3 domain of DNBP-1 (in pink) as displayed in ribbon format. The PTPN-22 proline-rich region is highlighted in green with prolines shown. Colored lines indicate predicted interactions between PTPN-22 and DNBP-1 within 6 Å. Predicted aligned error (PAE) plots of two of the highest-scoring models (rank_1 and rank_2) of PTPN-22 with the SH3.3 domain of DNBP-1. Yellow arrows indicate the region corresponding to the predicted interaction; green arrow indicates the PTPN-22 proline-rich region. ipTM scores for 10 different models (two seeds with six recycles) generated by AlphaFold-multimer were plotted for the indicated domains of DNBP-1 with full-length wild-type PTPN-22 or PTPN-22 containing a mutated PR.1 domain (PR.1 mut). Error bars represent 95% confidence intervals; ****p < 0.0001 based on a t test. (F) Confocal images showing young adults expressing CRISPR-generated PTPN-22::EGFP (green) and DNBP-1::mScarlet (magenta) in the region of the apical epidermis including inset (highlighted in yellow box.) The orange arrow indicates an example overlap (white) between PTPN-22::EGFP and DNBP-1::mScarlet. Sequences for *dnbp-1* mutant alleles can be found in the S1 File. Raw data are available in the S7 File.

contains a 59-bp deletion in exon 16, which leads to a frameshift that introduces seven novel amino acids after I915 followed by multiple stop codons (S1 File). Both CRISPR alleles of *dnbp-1* led to ~20% suppression of molting defects in *nekl-2*; *nekl-3* mutants (Fig 3B). RNAi-mediated knockdown of DNBP-1 using dsRNA injection methods led to ~40% suppression of molting defects in *nekl-2*; *nekl-3* mutants (Fig 3C), confirming that it is loss of DNBP-1 activity that confers genetic suppression. The observed difference in suppression levels between the *dnbp-1* genetic mutants and *dnbp-1(RNAi)* may be due in part to a pronounced reduction in fitness caused by the *dnbp-1* deletion mutation. We note that *dnbp-1(RNAi)* failed to suppress molting defects in moderate loss-of-function alleles of *nekl-2*(Y84L, G87A) and *nekl-3*(P194L) (Fig 3C). Together these results identify *dnbp-1* as a novel suppressor of *nekl-2*; *nekl-3* mutants and underscore the value of proximity labeling for identifying functionally relevant partners for proteins of interest.

We next took an in silico approach (AlphaFold-multimer; ColabFold) to better understand how DNBP-1 and PTPN-22 might physically interact [69,70]. The predicted structure of DNBP-1A isoform (DNBP-1.a) reveals two closely spaced SH3 domains at its N terminus, a Dbl-homology (DH) domain, a membrane-binding BAR domain, and a third SH3 domain at its C terminus (Fig 3D). As SH3 domains recognize proline-rich motifs, we used AlphaFold-multimer to predict interactions between DNBP-1 and PTPN-22 using full-length PTPN-22 and individual SH3 domains of DNBP-1 (SH3.1, SH3.2, and SH3.3). According to AlphaFold-multimer predictions, all 10 models suggested strong interactions between the N-terminal proline-rich region (aa 26–35; PPPPLPTSNP) of PTPN-22 and SH3.3 of DNBP-1 as evidenced by a mean ipTM (for interface predicted Template Modeling) score of 0.79 (Standard deviation, 0.062; range, 0.71–0.86) (Fig 3E). In contrast, the predicted models suggested a weak interaction or no interactions between the SH3.1 and SH3.2 domains of DNBP-1 with PTPN-22. Notably, these findings aligned closely with the yeast two-hybrid screen, which previously identified a physical association specifically between the SH3.3 domain of DNBP-1 and PTPN-22 [66]. Along these lines, both yeast two-hybrid experiments and protein modeling predicted an interaction between dynamin (DYN-1) and the SH3.1 domain of DNBP-1 (S4A Fig) [66]. We note that the failure of AlphaFold-multimer to predict an interaction between full-length DNBP-1 and PTPN-22 is not unexpected given the relative lack of confident structural predictions outside of the three SH3 domains (Fig 3E).

We further carried out co-localization experiments with transgenic animals that expressed CRISPR (endogenously) tagged PTPN-22::EGFP and DNBP-1::mScarlet. In the epidermis, PTPN-22::EGFP displayed both diffuse and punctate modes of expression, as well as a stripe-like assembly pattern along the portion of the apical epidermal surface that overlies the body wall muscles (Fig 3F). Consistent with an interaction between PTPN-22 and DNBP-1, we observed some correlation and overlap between the stripe patterns detected using PTPN-22::EGFP and the punctate structures identified with DNBP-1::mScarlet (Figs 3F and S4B). Moreover, the extent of PTPN-22 and DNBP-1 co-localization appeared consistent at different stages of development and in worms undergoing molting, although our analysis may have failed to detect subtle or highly transient stage-specific differences. DNBP-1::mScarlet also exhibited a pattern of mid- to large-sized puncta throughout the epidermis, which are reminiscent of endocytic compartments that co-localize with NEKL kinases as well as with epidermal CDC-42 (Figs 3F and S4B) [39,42]. We note that relatively weak expression of PTPN-22::EGFP in other regions of the epidermis made it difficult to infer the full extent of co-localization between these proteins (Fig 3F).

Given indications of a functional interaction between PTPN-22 and DNBP-1, we tested for genetic interactions between PTPN-22 and DNBP-1 by assaying for potential enhancement of *nekl-2*; *nekl-3* suppression when both PTPN-22 and DNBP-1 were simultaneously inhibited.

Our results suggested a significant increase in the percentage of suppressed *nekl-2*; *nekl-3* worms when both PTPN-22 and DNBP-1 were inhibited as compared with the loss of single proteins including putative null mutations (S4C Fig). Overall, our genetic results suggest that PTPN-22 and DNBP-1 are unlikely to be fully dependent on each other for their activities and also suggest that PTPN-22 may affect molting through one or more additional targets. Consistent with at least some functional independence, alteration of the N-terminal proline-rich domain of PTPN-22 (PR.1; $P_{26}$PPPLPTSNP$_{35}$ to AAAALATSNA, referred to as PR.1 mut), which is predicted to disrupt binding to the SH3.3 domain of DNBP-1, did not lead to strong suppression of *nekl-2; nekl-3* mutants (Figs 3E and S4D). Interestingly, although AlphaFold-multimer predicts that the $P_{26}$PPPLPTSNP$_{35}$ to AAAALATSNA alteration would preclude the binding of the DNBP-1 SH3.3 domain to this sequence, it also predicts that in the absence of the N-terminal proline rich domain, SH3.3 may instead bind to a C-terminal proline rich region in PTPN-22 (PR.2), thereby possibly preserving a functional interaction (Figs 3E and S4E). Collectively, our results indicate a physical and functional connection between DNBP-1 and PTPN-22, although the regulatory and functional consequences of this interaction remain to be determined.

## PTPN-22 participates in cell attachment and cytoskeletal regulation

Endogenously tagged PTPN-22::EGFP exhibited an intermittent stripe-like pattern at the apical surface of hyp7 in the region that overlies body wall muscles (Figs 3F, 4B, and S2 and S1 and S2 Movies). This expression pattern is characteristic of protein components that act in epidermal–ECM adhesive complexes termed *C. elegans* hemidesmosomes (CeHDs). CeHDs function to connect body wall muscles to the external cuticle, allowing for movement and embryonic morphogenesis [71–73]. Two epidermal CeHD components related to human

CeHDs and interact with the cuticular matrix [74,75]. Another epidermal transmembrane CeHD component, LET-805, is related to human Tenascins and localizes to the basal end of CeHDs where it interacts with the basement membrane that overlies muscle cells [76]. Muscle cells, in turn, bind to the overlying basement membrane through integrins (PAT-2 and PAT-3) [77–79]. Both MUP-4–MUA-3 and LET-805 bind to VAB-10, a dystonin family member, which in turn associates with intermediate filaments (IFs), which physically link the apical and basal complexes (Fig 4A) [74–76,80–83]. Interestingly, $P_{hyp7}$::PTPN-22::TurboID proximity labeling studies indicated enrichment of several CeHD proteins including, LET-805 (~2.6-fold), VAB-10 (~5.6-fold), PAT-12 (~6.0-fold), and IFB-1 (~3.4-fold) (Fig 4G and S4 File).

To determine if PTPN-22 co-localizes with CeHDs, we generated a strain expressing endogenously tagged PTPN-22::mScarlet, which resulted in an expression pattern that closely resembled that of PTPN-22::EGFP (Figs 3F and 4B). We then tested for co-localization with (non-CRISPR) GFP-tagged markers of CeHDs including MUP-4::GFP (apical CeHD), VAB-10A::GFP (apical and basal CeHD), IFB-1::GFP (intermediate filament), and LET-805 (basal CeHD). Our co-localization studies showed substantial overlap and/or adjacent localization between PTPN-22::mScarlet stripes and puncta with MUP-4::GFP, VAB-10A::GFP, IFB-1::GFP, and LET-805 (Fig 4B). However, the resolution of confocal microscopy was insufficient to determine the precise location of PTPN-22 in relation to apical or basal CeHD components.

To determine if PTPN-22 affects the function of CeHDs, we carried out partial/weak RNAi knockdown of several CeHD components in wild type and *ptpn-22* mutants and assayed for developmental defects. Specifically, we used RNAi feeding in conjunction with bacterial dilution to achieve different levels of target knockdown (see Materials and Methods) and looked for increased RNAi sensitivity in *ptpn-22* mutants. We note that knockdown of CeHD proteins

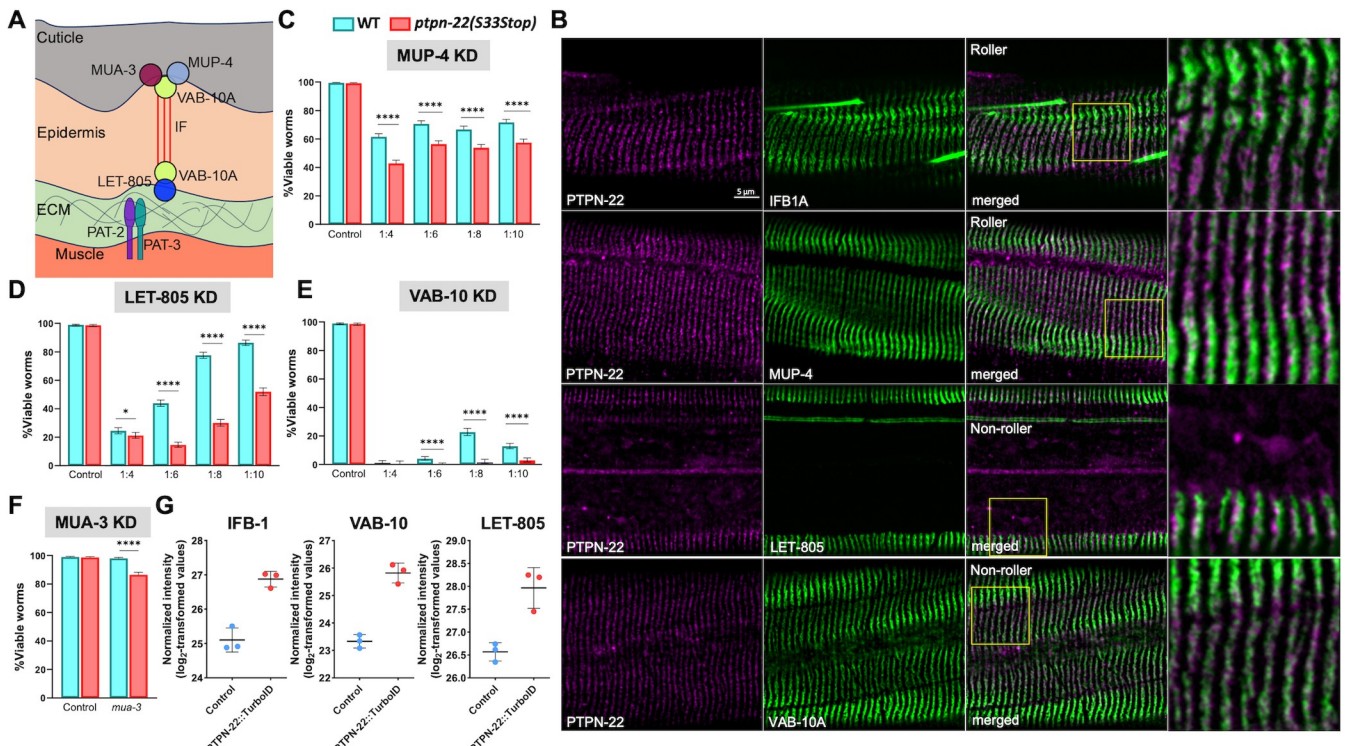

**Fig 4. PTPN-22 is localized to hemidesmosomes and shows a genetic interaction with CeHD proteins.** (A) Cartoon illustration depicting apical (MUA-3, MUP-4, and VAB-10A) and basal (LET-805 and VAB-10A) CeHD structural components within the epidermis. Intermediate filaments (IFs) connecting the complexes are indicated by red lines. The relative sizes of the different layers are not drawn to scale. Muscle cells attach to the basal lamina (extracellular matrix, ECM) separating the muscle and epidermis via α and β integrins (PAT-2 and PAT-3, respectively). (B) Co-localization in transgenic worms expressing endogenously tagged PTPN-22::mScarlet and GFP-tagged CeHD proteins (IFB-1A::GFP, MUP-4::GFP, LET-805::GFP, and VAB-10A::GFP). Note that PTPN-22::mScarlet; IFB-1A::GFP and PTPN-22::mScarlet; MUP-4::GFP transgenic worms exhibited a rolling (twisted) phenotype because of the presence of dominant *rol-6* (su1006) transgene in these backgrounds (see S6 File). (C–F) RNAi feeding knockdown (KD) of *mup-4* (C), *let-805* (D), *vab-10* (E), and *mua-3* (F) was carried out in wild-type and *ptpn-22(S33Stop)* worms using the indicated dilution series. Error bars represent 95% confidence intervals. Fisher's exact test was used to calculate p-values; ****p < 0.0001; *p < 0.05. (G) Dot plots show the enrichment of CeHD proteins in the P_hyp7::PTPN-22::TurboID samples. Error bars represent standard deviation. Raw data are available in the S7 File.

can lead to either detachment of the cuticle from the epidermis or detachment of the muscle from the epidermis, both of which lead to larval arrest and lethality [72,74–76,80–84]. We observed that loss of function of *ptpn-22* significantly enhanced larval lethality caused by *mup-4(RNAi)* at multiple dilutions (Fig 4C). Likewise, *ptpn-22* was more sensitive to *let-805(RNAi)* at all dilutions, leading to an ~3-fold increase in larval lethality at 1:6 and 1:8 dilutions (Fig 4D). In the case of *vab-10(RNAi)*, we observed significant embryonic lethality at all dilutions, although a substantial increase in embryonic arrest was observed for *ptpn-22* at 1:8 (~16-fold) and 1:10 (~5-fold) dilutions versus N2 (Fig 4E). Finally, we observed a small but statistically significant increase in larval lethality of *ptpn-22* treated with (non-diluted) *mua-3 (RNAi)* versus N2 (Fig 4F).

One possible explanation for the observed RNAi-enhancement of hemidesmosome genes is that loss of *ptpn-22* leads to non-specific hypersensitivity to RNAi, which has been observed for mutations in several genes including *lin-35* and *rrf-3* [85–88]. We therefore examined RNAi-feeding phenotypes for five genes (*rsr-2, unc-87, tir-1, mom-2,* and *hmr-1*) previously reported to show enhanced defects in *lin-35* and/or *rrf-3* backgrounds. Whereas *lin-35* and *rrf-3* mutants showed increased RNAi sensitivity to these five genes relative to N2 controls, *ptpn-22* mutants failed to exhibit a consistent pattern of hypersensitivity (S5 Fig). Taken together,

our genetic and localization data suggest that PTPN-22 plays a positive role in the function of *C. elegans* hemidesmosomes. Nevertheless, we failed to detect gross changes in the localization pattern of CeHD components (MUP-4, IFB-1A, and LET-805) in *ptpn-22* null mutants (S6A Fig), consistent with the viability and normal morphology of *ptpn-22* mutants. As such, PTPN-22 could play a role in fine-tuning CeHD function, assembly, or stability.

Given that CeHDs are extensively remodeled during molting cycles, we were curious if loss of *ptpn-22* might contribute to *nekl* molting suppression in part through its effects on CeHDs in addition to its implicated interactions with DNBP-1. As a test for this, we carried out partial RNAi knockdown of the hemidesmosome proteins *mup-4* and *let-805* in *nekl-2; nekl-3* worms but failed to observe any mitigation of the molting-defective phenotype (S6B Fig). These results suggest that the role of PTPN-22 at CeHDs may be distinct from functions linked to the NEKL–MLT pathway.

Previous studies have indicated the involvement of mammalian PTPN22 in suppressing T lymphocyte–specific integrin activation to regulate integrin-mediated cell adhesion [31,32]. Correspondingly, $P_{hyp7}$::PTPN-22::TurboID proximity labeling studies showed enrichment of the α subunit of integrin PAT-2/ITGA2B/ITGA5 (~5-fold) and epithelial junction protein DLG-1/DLG1 (~4.3-fold) (Figs 5A and S7A and S4 File) [78,89]. To explore a potential role for *C. elegans* PTPN-22 in integrin function and other adherens junctions beyond CeHDs, we conducted RNAi enhancement tests to assess *ptpn-22* genetic interactions with *pat-2* and *dlg-1*. *ptpn-22(S33Stop)* worms had significantly higher paralysis rates when subjected to various dilutions of *pat-2(RNAi)* feeding treatment as compared with wild-type worms (Fig 5B and 5C). Interestingly, the combination of *ptpn-22* loss of function and *pat-2(RNAi)* also resulted in a significant decrease in the size of the worms, as assessed by body length measurements (Fig 5C). Additionally, upon *dlg-1(RNAi)* treatment, *ptpn-22(S33Stop)* worms showed enhanced embryonic lethality (S7B and S7C Fig). Taken together, these findings suggest that PTPN-22 plays roles in multiple adhesive structures in the epidermis and, potentially, in the muscle.

Our proximity labeling data also detected enrichment of multiple actin regulatory proteins including the conserved actin-capping proteins CAP-1/CAPZA1/2, CAP-2/CAPZB, and GSNL-1/CAPG. CAP-1 and CAP-2 were among the 29 proteins that exhibited enrichment in both PTPN-22::TurboID samples (~4-fold and ~3-fold, respectively) and $P_{hyp7}$::PTPN-22::TurboID samples (~6-fold and ~7-fold, respectively) (Figs 5A and S3E). GSNL-1 was enriched ~3.2-fold in the $P_{hyp7}$::PTPN-22::TurboID samples (S7A Fig). Actin-capping proteins maintain the ratio between globular monomeric actin (G-actin) and filamentous actin (F-actin) by capping the growing end of actin fibers and have roles in embryonic development and tissue morphogenesis [90–95]. Using our RNAi enhancement approach, we observed that *ptpn-22 (S33Stop)* mutants showed higher embryonic lethality after *cap-2(RNAi)* than wild-type worms (Fig 5B and 5D). In addition, hatched *ptpn-2(S33Stop); cap-2(RNAi)* worms exhibited uniform early larval arrest, which was not observed in wild-type worms treated with *cap-2(RNAi)* (Fig 5B). In contrast, no observable phenotypic defects were detected in either wild-type or *ptpn-22 (S33Stop)* worms when subjected to *gsnl-1(RNAi)* treatment by feeding (S7D Fig). Together, these results implicate PTPN-22 in the regulation of cell adhesion and the actin cytoskeleton during development.

## Discussion

In this study we identified PTPN-22, a conserved tyrosine phosphatase non-receptor type, as an effector of *C. elegans* molting. Specifically, loss of PTPN-22 catalytic activity alleviated molting defects in *nekl* mutant backgrounds with partial-to-moderate loss of function. An

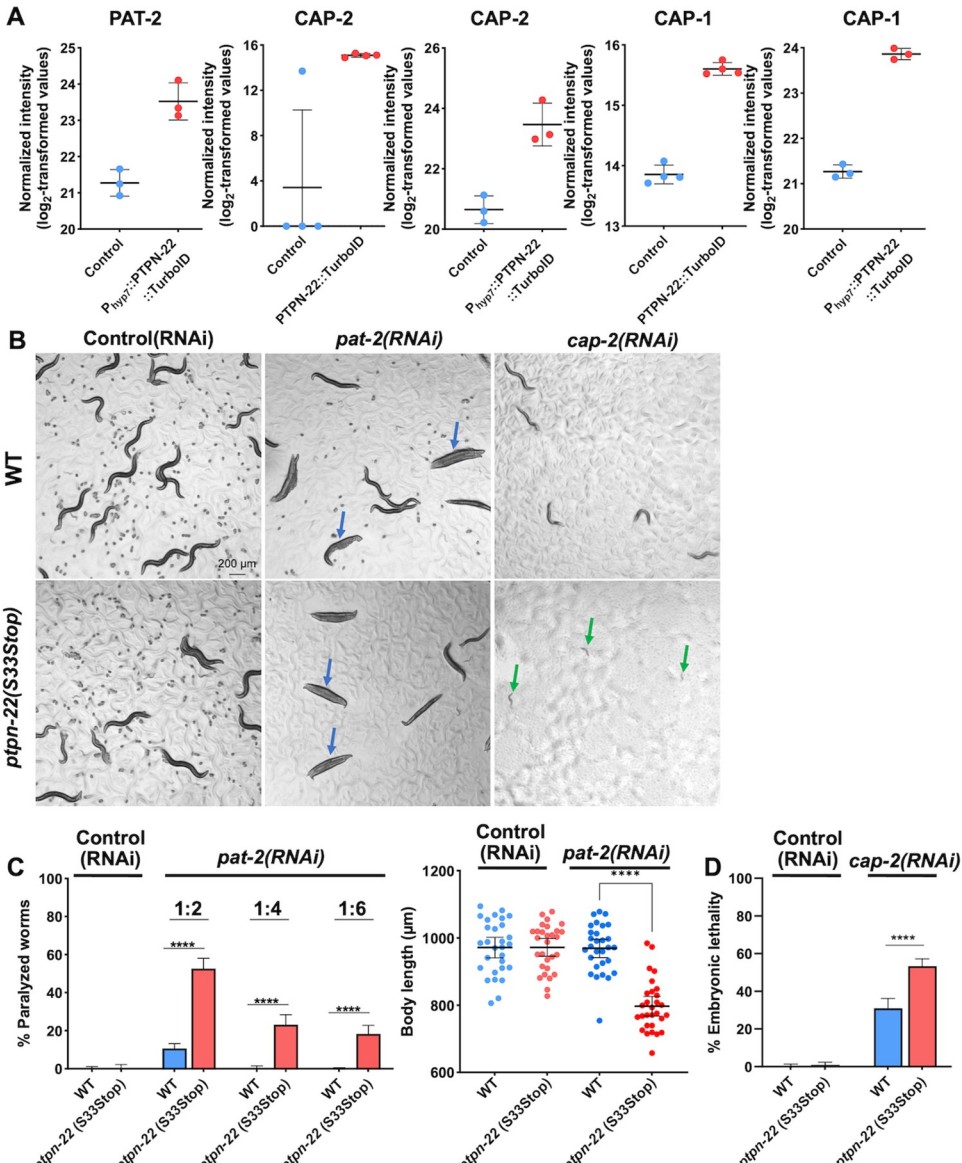

**Fig 5. PTPN-22 interactions with cell attachment and actin regulatory proteins.** (A) Dot plots showing the enrichment of proteins in the PTPN-22::TurboID and $P_{hyp7}$::PTPN-22::TurboID samples. Error bars represent standard deviation. (B) Bright-field images of wild-type and *ptpn-22(S33Stop)* worms on control (empty vector) or *pat-2* or *cap-2* RNAi feeding plates. Blue arrows indicate paralyzed adults; green arrows indicate arrested L1 larvae. (C) Bar graphs show the percentage of paralyzed worms in the indicated RNAi feeding experiments. Fisher's exact test was used to calculate p-values. Dot plot shows body length measurement of individual worms of the indicated backgrounds on control (empty vector) and *pat-2* RNAi feeding plates. Statistical significance was determined using a two-tailed, unpaired t-test. Error bars represent 95% confidence intervals. (D) Bar graphs show the percentage of embryonic lethality in the indicated RNAi feeding experiments. Fisher's exact test was used to calculate p-values. Error bars represent 95% confidence intervals. ****p < 0.0001. Raw data are available in the S7 File.

appealing model is that the NEKLs and PTPN-22 may exert opposing effects on one or more common substrates or may act on distinct components within a discrete complex or pathway. Along those lines, we previously identified a mutation affecting PAA-1/PPP2R1, a conserved serine/threonine PP2A phosphatase subunit, as a suppressor of *nekl* defects [41]. Nevertheless, whereas PP2A could directly reverse NEKL-mediated phosphorylation at serines and

threonines, PTPN-22 is predicted to target specifically phosphotyrosines and may thus oppose NEKL functions more indirectly.

We previously showed that NEKL-2 and NEKL-3 have overlapping but distinct roles in several steps of membrane trafficking including cargo uptake from apical and basolateral membranes along with transit through the endosomal system [38,40]. Additionally, we showed that the NEKLs inhibit actin filamentation, co-localize at endosomes with CDC-42, and negatively regulate CDC-42 activity [42]. CDC42 is a well-studied member of the Rho family of GTPases that promotes actin polymerization within a variety of cellular contexts including multiple roles within the membrane trafficking system. Effectors of CDC42 include conserved members of the WASP and TOCA family of proteins, which together with the Arp2/3 complex promote the extension and branching of actin filaments [96–100]. CDC42 activity is itself tightly controlled by upstream regulators including RhoGEFs, such as DNBP-1 family proteins, as well as GTPase-activating proteins (GAPs). DNMBP, the human ortholog of DNBP-1, regulates actin assembly by serving as a scaffold for CDC42 and WASP family proteins and also binds to dynamin, a membrane-remodeling enzyme that promotes vesicle fission [67,68,101]. Our findings from proximity labeling, molecular modeling, and subcellular localization studies, in conjunction with previously published yeast two-hybrid interaction data, suggest that PTPN-22 may be an accessory component or regulator of this complex [66]. Such a model is consistent with our current and previously published genetic data showing that inhibition of PTPN-22, DNBP-1, CDC-42, and SID-3/ TNK2/ACK1 can suppress *nekl* molting defects. Nevertheless, the precise role of PTPN-22 in this context remains to be determined.

Further examples of the potential involvement of PTPN-22 in actin regulation come from our proximity labeling experiments, which suggested an interaction between PTPN-22 and several actin capping proteins (CAP-1, CAP-2, and GSNL-1), each of which blocks actin filament elongation. Additionally, our genetic experiments revealed a robust genetic interaction between *cap-2* and *ptpn-22*, as loss of function of both proteins led to a significant increase in embryonic lethality and developmental arrest at early larval stages. Together these findings further implicate PTPN-22 in the regulation of actin polymerization.

PTPN-22 proximity labeling studies also identified several CeHD components (IFB-1, PAT-12, VAB-10, and LET-805), a finding strongly supported by the observed co-localization between PTPN-22 and IFB-1A, MUP-4, VAB-10, and LET-805. These results, together with RNAi enhancement studies showing that *ptpn-22* mutants are hypersensitive to partial knockdown of CeHD proteins, suggest a role for PTPN-22 in positively regulating the function of CeHD proteins. Additionally, proximity labeling and genetic interaction studies implicated a connection between PTPN-22 and PAT-2, a component of the integrin-attachment complex, and DLG-1, an epithelial junctional protein [72,78,89]. As noted in the results, however, our proximity labeling studies are likely to contain at least some false (non-specific) positives, which could include highly abundant proteins such as those acting within cell adhesion complexes. Nevertheless, by coupling proximity labeling to localization data and functional/genetic studies, our combined results indicate functions for PTPN-22 in the regulation of cell attachments and serve as a basis for future studies. Moreover, previous work on human PTPN22 suggests that this could be a conserved function given that mammalian PTPN22 co-localizes with a T cell–specific integrin, regulates its activity, and affects lymphocyte adhesion [31].

Proximity labeling and PTPN-22 expression studies also implicated PTPN-22 in germline functions through an apparent connection to P granules. P granules are well-studied RNP-based membraneless organelles located in the perinuclear region of germline nuclei, where they act to restrict mRNA cytoplasmic import and protein translation [64,102–104]. Most notably, nearly one-third of the 179 proteins identified by PTPN-22::TurboID are known P granule components (S2 File; Sheet "Overlap with P granules"). Our results also indicated

significant enrichment of nuclear transport proteins, particularly nuclear pore complex components (NPP-1, NPP-4, NPP-9, and NPP-14) as well as importins (IMA-2 and IMA-3), in the PTPN-22::TurboID samples (S2 File). P granules are located directly adjacent to nuclear pore complexes, thereby enabling the rapid sequestration of mRNAs as they pass through nuclear pores [103,105]. Additionally, it is possible that an interaction between PTPN-22, nuclear pore complex proteins, and importins could occur in other tissues, as we observed expression of PTPN-22 in nuclear compartments of other cell types including the vulva, pharynx, intestine, and seam cells. Previous studies on the subcellular localization of human PTPN22 demonstrated expression within the cytoplasm and nucleus of macrophages and reported distinct functional roles for nuclear and cytoplasmic PTPN22 [106]. Nonetheless, the functions of PTPN22 in nuclear and perinuclear compartments are poorly understood.

In summary, our research highlights the value of combining the complementary approaches of forward genetics and proximity labeling to gain novel insights into gene functions. More specifically, our studies substantially expand upon the known functions of PTPN22 family members beyond the adaptive immune system, including evidence for new molecular, cellular, and developmental functions.

## Materials and methods

### Strain maintenance

*C. elegans* strains were maintained using established protocols and cultured at 22˚C, unless otherwise indicated. The strains used in this study are listed in the S6 File.

### Genome editing

Established CRISPR/Cas9 approaches were used for genome editing in *C. elegans* [107]. Ape, CHOPCHOP, and CRISPRcruncher were used for designing guide RNAs and repair templates [108–112].

### RNAi

dsRNAs corresponding to *ptpn-22* and *dnbp-1* were generated according to standard protocols [113]. dsRNA at a concentration of 500–800 ng/µL was injected into the gonads of day-1 adult worms. The RNAi feeding protocol was adapted from the procedure outlined by Conte et al. [114]. Briefly, six L4-stage worms were transferred to experimental RNAi plates and transferred every 24 hours for two more times onto fresh RNAi plates. Phenotypic defects were assessed ~72 hours after their initial placement. To quantify embryonic lethality, eggs were allowed to hatch for 24 hours, after which dead embryos were counted. For the genes *mup-4*, *let-805*, *vab-10*, and *pat-2* (as shown in Figs 4 and 5), the penetrance of the RNAi phenotypes was reduced by diluting the bacterial culture with the control (empty vector) RNAi bacterial culture at different ratios. The RNAi clones used were obtained from the Ahringer library [115].

### Pulldown of biotinylated proteins

The protocol for the pulldown of biotinylated proteins was adapted from the protocol described by Sanchez et al. [52]. Briefly, mixed-stage worms were cultured at 22˚C (except for S3 Fig, where worms were cultured at 25˚C). Subsequently, the worms were washed off the plates with ddH$_2$O and washed multiple times with ddH$_2$O to eliminate bacteria. Excess water was carefully removed. One volume of RIPA lysis buffer supplemented with protease inhibitors (Thermo Fisher, Cat# 78442) was added to the worm pellet. The pellet was then

homogenized using a handheld homogenizer (Huanyu MT-13K-L) for 4 minutes until complete lysis was achieved, and intact worms were no longer visible. The lysate was then centrifuged twice at 14,000× *g* for 8 minutes, and the supernatant was transferred to a fresh tube after each spin. Streptavidin-coated magnetic beads (Thermo Fisher, Cat# 65001) were equilibrated by washing them twice with RIPA lysis buffer. The lysate was added to the beads and gently rotated on a spinning wheel at room temperature for 1 hour (except for S3 Fig, where the lysate was mixed with beads at 4˚C for 16 hours). The beads were separated from the lysate using a magnetic stand. The beads were then washed with five buffers to eliminate non-specifically bound proteins as follows: 1× wash with cold RIPA lysis buffer, 1× wash with cold 1 M KCl, 1× wash with cold 0.1 M $Na_2CO_3$, 1× wash with cold 2 M urea in 10 mM Tris-HCl (pH 8.0), 3× washes with cold RIPA lysis buffer, and 5× washes with cold 1× PBS. The beads were snap-frozen and stored at –80˚C until ready for on-bead digestion.

## Western blot analysis

Worms were lysed in RIPA buffer containing Tris-HCl (pH 8.0) 50 mM, NaCl 150 mM, SDS 0.2%, Sodium deoxycholate 0.5%, Triton X-100 1%, and Halt Protease and Phosphatase Inhibitor Single-Use Cocktail 1× (Thermo Fisher, Cat# 78442). Proteins from the lysates were separated by SDS-PAGE, transferred to nitrocellulose filters, and immunoblotted with antibodies using standard methods. Horseradish peroxidase(HRP)–conjugated streptavidin (streptavidin-HRP; Cat# 3999S) and HRP-conjugated rabbit monoclonal antibody against β-actin (Cat# 5125S) were obtained from Cell Signaling Technology. HRP-conjugated mouse monoclonal antibody against FLAG was obtained from GenScript (Cat# A01428).

To detect biotinylated proteins by western blot methods, 10 μl of beads were mixed with 3× Laemmli SDS sample buffer and 2 mM biotin, and then boiled at 95˚C for 10 minutes. Beads were pelleted using a magnetic stand and the supernatant was collected for subsequent use in western blot analysis.

## Mass spectrometry data analysis

Protein samples were reduced, alkylated, and digested on-bead using filter-aided sample preparation with sequencing-grade modified porcine trypsin (Promega) [116]. Briefly, tryptic peptides were then separated by reversed-phase XSelect CSH C18 2.5-μm resin (Waters) on an in-line 150 x 0.075–mm column using an UltiMate 3000 RSLCnano system (Thermo Fisher). Peptides were eluted using a 60-minute gradient of buffer A/buffer B from 98:2 to 65:35. Eluted peptides were ionized by electrospray (2.2 kV) followed by mass spectrometric analysis on an Orbitrap Exploris 480 mass spectrometer (Thermo Fisher). After data acquisition, data were searched using an empirically corrected library against the UniProt *Caenorhabditis elegans* database and a quantitative analysis was performed to obtain a comprehensive proteomic profile. Spectronaut (Biognosys version 18.5) was used for database search using the directDIA method (Fig 2). Notably, for S3 Fig, proteins were identified and quantified using Encyclope-DIA and visualized with Scaffold DIA (Proteome Software, Portland, Oregon, USA) using a 1% false discovery threshold at both the protein and peptide level [117]. Protein MS2 exclusive intensity values were assessed for quality using ProteiNorm [118]. The data were normalized using VSN (Fig 2) [119] or cyclic loess (S3 Fig) [120], and were analyzed using proteoDA [121] (111) to perform statistical analysis using Linear Models for Microarray Data (limma) with empirical Bayes (eBayes) smoothing to the standard errors [120–122].

To estimate the proportion of the four biotinylated carboxylases (MCCC-1, PCCA-1, PYC-1, and POD-2) in the MS samples we first averaged their individual abundance from the technical replicates (e.g., a1+a2+a3+a4/4 = aa), then converted these values from log2 scale (2^aa),

and then summed these four values (carboxylase-total). Likewise, we summed the values for all detected proteins in the sample (all-total). Lastly, we determined the percentage of the carboxylases in the total sample [(carboxylase-total/all-total)*100].

## Image acquisition and image processing

All confocal images were acquired using an Olympus IX83 inverted microscope with a Yokogawa spinning-disc confocal head. z-Stack images were acquired using a 100×, 1.35 N.A. silicone oil objective. cellSense 3.3 software (Olympus Corporation) was used for image acquisition.

For colocalization studies, the raw z-stack images were deconvoluted using the Wiener deconvolution algorithm (cellSense 3.3 software). The desired z-plane was extracted from the deconvoluted z-stack images for further analysis.

## Statistics

All statistical tests were performed using software from GraphPad Prism following standard procedures [123].

## Supporting information

**S1 Fig. Multiple sequence alignment of *C. elegans* PTPN-22 and its orthologs in other organisms.** Jalview was used to visualize multi-sequence alignments among *C. elegans* PTPN-22, human PTPN12 and PTPN22, mouse PTPN22 and PTPN18, chicken PTPN12 and PTPN22, and zebrafish PTPN12 and PTPN22. Conserved residues, based on sequence homology, are highlighted in purple. A consensus sequence is provided below the sequence alignments.
(PDF)

**S2 Fig. PTPN-22 expression during development.** (A) A confocal microscopy image of the anterior body of a day-1 adult worm expressing PTPN-22::GFP. White arrows show examples of PTPN-22::EGFP in pharyngeal cells; white arrowheads show its expression in nuclear compartments of pharyngeal cells; yellow arrows indicate autofluorescence of the cuticle. (B) An L2 larva expressing PTPN-22::EGFP. White arrowheads indicate seams cells, which show expression in their cytoplasm and nuclei; red arrows indicate nerve cord cells; yellow arrows indicate autofluorescence of the cuticle. (C) Expression of PTPN-22::EGFP in the germline of an L4-stage worm including nuclear and perinuclear expression as indicated with red arrowheads; blue arrowhead indicates a distal tip cell; blue bracket indicates spermatheca; yellow arrows indicate autofluorescence of cuticle. Yellow box corresponds to the enlarged inset, which shows expression of PTPN-22::EGFP in the perinuclear region of germline nuclei. (D) Ubiquitous nuclear and cytoplasmic expression of PTPN-22::mScarlet in early embryonic cells (left), with the DIC (middle) and merged (right) images. (E) Nuclear and cytoplasmic expression of PTPN-22::mScarlet in vulval cells in an L4-stage worm (white arrows) with the DIC (middle) and merged (right) images. The green arrows show expression in intestinal nuclei; blue arrows show gut granule autofluorescence; white bracket indicates proximal somatic gonad cells.
(PNG)

**S3 Fig. Results of P_hyp7::PTPN-22::TurboID studies.** (A) Schematic showing relevant portion of the expression construct used to drive PTPN-22::TurboID in the major hyp7 epidermal syncytium. A *ptpn-22* cDNA was fused to coding sequences for TurboID and a 3×FLAG tag (both placed at the C terminus) and expressed under the control of a hyp7-specific (Y37A1B.5)

promoter. Western blot images of representative N2 and $P_{hyp7}$::PTPN-22::TurboID samples show the biotinylated proteins in the input and pulldown (IP) fraction after blotting with streptavidin-HRP. The lower two blots on the left show $P_{hyp7}$::PTPN-22::TurboID expression based on an antibody against FLAG (upper) and a loading control with an antibody against β-actin (lower). (B) The volcano plot shows the enrichment of proteins after LC-MS/MS analysis in the $P_{hyp7}$::PTPN-22::TurboID samples (red) and in the N2 samples (blue). (C) The dot plot shows normalized intensity values of ectopically expressed PTPN-22cd::TurboID::3×FLAG versus the N2 control (three replicates each). Error bars represent standard deviation. (D, E) Venn diagram shows the overlap of enriched proteins between $P_{hyp7}$::PTPN-22::TurboID samples and P granule proteins (D) and the overlap of enriched proteins between PTPN-22::TurboID samples and $P_{hyp7}$::PTPN-22::TurboID samples (E) (see S4 File). cd, cDNA; STV, streptavidin.
(PNG)

**S4 Fig. DNBP-1 interactions and expression.** (A) ipTM scores for five different models generated using AlphaFold-multimer, each of which was used to determine predicted binding between the three SH3 domains (SH3.1, SH3.2, and SH3.3) of DNBP-1 and two isoforms of DYN-1 (DYN-1.a and DYN-1.b). Error bars represent standard deviation. PAE plots of the best models for DYN-1.a and DYN-1b interactions with DNBP-1(SH3.1) are shown with green arrows indicating the predicted proline-rich region of interaction at the C terminus of DYN-1.a and DYN-1.b. (B) Representative confocal images of day-1 adult worms expressing DNBP-1::mScarlet in the anterior and posterior side of the epidermis. (C,D) Bar graph showing the percentage of suppressed worms in the indicated backgrounds. Error bars represent 95% confidence intervals. Fisher's exact test was used to calculate p-values; ****$p < 0.0001$; ns, not significant. (E) PAE plots showing the two highest-scoring Alphafold2 multimer interactions models (rank_1 and rank_2) of PTPN-22(PR.1mut) with the SH3.3 domain of DNBP-1. Yellow arrows indicate the predicted interacting region. Raw data are available in S7 File.
(PNG)

**S5 Fig. RNAi feeding sensitivity assessment in N2, *ptpn-22*, *rrf-3*, and *lin-35* backgrounds.** (A–E) $P_0$ L4 larval of the indicated genotypes were placed on control *GFP(RNAi)* or experimental (*rsr-2*, *unc-87*, *mom-2*, *tir-1*, or *hmr-1*) RNAi-feeding plates for 24 h, and then moved to new RNAi-feeding plates and allowed to lay eggs for ~8 h before removing the $P_0$ adults. $F_1$ progeny, were scored ~1–4 days following the removal of $P_0$ animals depending on the nature of the assessed phenotype. Raw data are available in the S7 File.
(PNG)

**S6 Fig. CeHD protein expression in wild type and *ptpn-22* mutants and genetic interactions of CeHD proteins with *nekl-2; nekl-3* mutants.** (A) Confocal microscopy images of day-1 adults of the indicated backgrounds expressing MUP-4::GFP, IFB1A::GFP, and LET-805::GFP. Note that no gross differences in the localization of CeHD proteins were detected. (B) RNAi-suppression experiments were carried out with wild-type and *nekl-2*; *nekl-3* mutants after partial knockdown of *mup-4* and *let-805* using RNAi feeding at different dilutions (see Materials and Methods). Note that no reduction in the percentage of *nekl-2; nekl-3* arrest was observed. Error bars represent 95% confidence intervals. Fisher's exact test was used to calculate p-values; ns, not significant. Raw data are available in the S7 File.
(PNG)

**S7 Fig. PTPN-22 interaction with PAT-2 and actin-capping proteins.** (A) Dot plots show the enrichment of the indicated proteins in the N2 and $P_{hyp7}$::PTPN-22::TurboID samples. (B) Bright-field images of wild-type and *ptpn-22(S33Stop)* worms on control (empty vector) or

*dlg-1* RNAi feeding plates. Red arrow shows the presence of dead eggs on the *dlg-1(RNAi)* plate. (C, D) Bar graphs show the percentage of embryonic lethality (C) and viable adults (D) of the indicated backgrounds in control (empty vector) and *gsnl-1* RNAi feeding plates. Raw data are available in S7 File.
(PNG)

**S1 File. CRISPR templates, primers and design information.**
(PDF)

**S2 File. Proteomic data and analyses for PTPN-22::TurboID studies.**
(XLS)

**S3 File. Gene ontology analyses of proteins identified in PTPN-22::TurboID studies.**
(XLSX)

**S4 File. Proteomic data and analyses for P$_{hyp7}$::PTPN-22::TurboID studies.**
(XLS)

**S5 File. Gene ontology analyses of proteins identified in P$_{hyp7}$::PTPN-22::TurboID studies.**
(XLSX)

**S6 File. Strain list.**
(XLSX)

**S7 File. Raw data for all figures and supplemental figures.**
(XLSX)

**S1 Movie. Z-stack movie of an L4 worm expressing PTPN-22::GFP.**
(AVI)

**S2 Movie. Timelapse movie of an L4 worm expressing PTPN-22::GFP in the hemidesmosome.**
(AVI)

## Acknowledgments

We thank Amy Fluet for editing this manuscript. We also thank Stephanie Byrum and Sam MacKintosh at the IDeA National Resource for Quantitative Proteomics for performing proteomic analyses and for helpful discussions, and Zachary Davis for technical assistance.

## Author Contributions

**Conceptualization:** Shaonil Binti, David S. Fay.

**Data curation:** Shaonil Binti, Adison G. Linder, Philip T. Edeen, David S. Fay.

**Formal analysis:** Shaonil Binti, Adison G. Linder, Philip T. Edeen, David S. Fay.

**Funding acquisition:** David S. Fay.

**Investigation:** Shaonil Binti, Adison G. Linder, Philip T. Edeen, David S. Fay.

**Methodology:** Shaonil Binti, Adison G. Linder, Philip T. Edeen, David S. Fay.

**Project administration:** David S. Fay.

**Resources:** David S. Fay.

**Supervision:** Shaonil Binti, David S. Fay.

**Validation:** Shaonil Binti, Adison G. Linder, David S. Fay.

**Visualization:** Shaonil Binti, Adison G. Linder, David S. Fay.

**Writing – original draft:** Shaonil Binti, David S. Fay.

**Writing – review & editing:** Shaonil Binti, David S. Fay.

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
