## [Decision Letter · Decision Letter 0]

13 May 2024

Dear Dr Fay,

Thank you very much for submitting your Research Article entitled 'A conserved protein tyrosine phosphatase, PTPN-22, functions in diverse developmental processes in C. elegans' to PLOS Genetics.

The manuscript was fully evaluated at the editorial level and by independent peer reviewers. The reviewers appreciated the attention to an important topic and overall they were enthusiastic. But, they identified some minor concerns that we ask you address in a revised manuscript. 

We therefore ask you to modify the manuscript according to the review recommendations. Your revisions should address the specific points made by each reviewer.

Yours sincerely,

Anne C. Hart

Academic Editor

PLOS Genetics

Gregory P. Copenhaver

Section Editor

PLOS Genetics

Reviewer's Responses to Questions

**Comments to the Authors:**

Reviewer #1: Summary

The NEKL-2 and NEKL-3 kinases are essential for molting and loss results in larval lethality as previously demonstrated by the Fay lab. While weak loss of function alleles of nekl-2 or nekl-3 are not lethal, but the nekl-2;3 double hypomorphic mutant is lethal. Here Binti et al report the identification of a mutation in ptpn-22 as a genetic suppressor screen the nekl-2;3 lethality. This is confirmed with several ptpn-22 alleles and RNAi. However, loss of ptpn-22 fails to suppress strong loss of function alleles in nekl-2 and nekl-3 like previously described suppressors. PTPN-22 is a Protein Tyrosine Phosphatase of the Non-receptor type. Mutation of the catalytic Cysteine by CRISPR also suppresses nekl-2;3 consistent with it functioning in the capacity of a phosphatase.

To better understand of how PTPN-22 function to suppress nekl-2;3 they performed BioID to identified proximity interactors in quadruplicate. PTPN-22 proximity interactors are enriched for pathway components involved in nucleocytoplasmic transport, spliceosome and and endocytosis among others. Strikingly, the PTPN-22 BioID interactome is very enriched in proteins that localize to P granules in the germline. PTPN-22 GFP and mScarlet fusions are widely expressed in many tissues including near germline P granules. Since NEKL-2 and NEKL-3 function in the hypodermis to regulate molting, an additional BioID was done using a strain that expresses specifically in the hypodermis. This set of experiments identified a number of hits but may have been less specific based on the high number of proteins enriched in the control.

DNBP-1 was among the hits enriched in the whole organism BioID and was previously found to interact with PTPN-22 in a large scale SH3 domain yeast two-hybrid screen. DNBP-1 is a Dynamin interacting protein. Two mutations were generated which as with RNAi knockdown suppress the nekl-2;3 mutant, but not stronger alleles as seen with ptpn-22 mutants. Consistent with the yeast two hybrid data, the third SH3 domain of DNBP-1 was predicted to interact with the N-terminal Proline rich region of PTPN-22. DNBP-1::mScarlet localizes in a punctate manner reminiscent of endocytic compartments that partially overlap with PTPN-22::GFP suggesting that they may interact in some regions of the hypodermis.

Although dnbp-1 and ptpn-22 loss both suppress nekl-2;3 larval lethality, the loss of both dnbp-1 and ptpn-22 had an additive effect indicating that although they physically interact, they may function by different means to regulate molting. Consistent with this, mutating that the Proline-rich region of PTPN-22 that is predicted to be required for DNBP-1 binding does not abrogate its function with regard to suppressing nekl-2;3.

PTPN-22 proximity interactors include components of hemidesmosomes (HDs) and PTPN-22 localizes with or near HDs. The lethality of HD component RNAi is enhanced by a ptpn-22 mutant suggesting that PTPN-22 could function with HDs. However, HD component localization does not appear to be perturbed in ptpn-22 mutants suggesting that it is not regulating localization. HD RNAi does not suppress nekl-2;3 lethality suggesting that they do not function with PTPN-22 in the molting pathway. Interestingly, ptpn-22 mutant enhanced RNAi phenotypes of other proximity interactors including PAT-2 integrin, DLG-1, and CAP-2.

Significance

In comparison to PTKs, PTPs are less studied. The PTPN22 protein is mostly studied in the context of immunology yet it is widely expressed. In C. elegans the ortholog PTPN-22 is virtuely uncharacterized. Binti et al describe a role for PTPN-22 in the regulation of molting in the context that it can suppress the lethality of a doubley hypomorphic nekl-2; nekl-3 mutant. They conduct two BioID screens with PTPN-22 identifying many proximity interactors that would be of interest to others in the field, particularly those interested in P granules. Among them they identify DNBP-1 for which they find also suppresses the lethality of nekl-2; nekl-3 mutants implicating it in the molting pathway as well. Although they do not appear to function together in this molting pathway, these findings suggest that PTPN22 may function with DNBP1 and dynamin in other contexts. Further they demonstrate potential roles for PTPN-22 in functioning with hemidesmosomes, integrin and actin capping proteins.

Overall, the paper is very well written, and the experiments are thorough and well controlled. The authors back up all their claims with data. The authors are candid about potential short comings of some of the BioID experiments. The raw data is provided in supplementary files.

I have only one major concern. Could the enhancement of RNAi phenotypes of the HD components, pat-2, dlg-1, and cap-2 in the ptpn-22 mutant be due the mutant ptpn-22 itself, or another mutation in the background resulting in an Eri (Enhancer of RnaI) phenotype. I ask this because PTPN-22 localizes near P granules and interacts with many P granule components that include genes involved in small RNA pathways. Thus, an alternative explanation for the RNAi data in figures 4 and 5 could be that ptpn-22 mutants are general enhancers of RNAi.

Minor comments/corrections

Line 160. “Transgenic animals carrying…” This is mutant strain, but it is not clear what the transgene is.

Line 205. It would be good to indicate that N2 is the wild type strain for clarity.

Line 319. DNBP-1 is miswritten as DBNP-1

Fig S3A The background for the FLAG immunoblot is very faint and appears to be the same as the one above, perhaps adding a line or more space between the blots to avoid confusion.

Reviewer #2: This is a very comprehensive paper, using many protein interaction and protein localization approaches to address the cell biological function of PTPN-2 phosphatase. The gene was discovered by genetics but the characterization is quite biochemical and cell biological, which is rare for an excellent C. elegans genetic analysis paper. The proximity labelling is carefully analyzed and explained and the cell biology analysis that also includes tissue-specific rescue and cell biology is outstanding. The initial genetics that identified ptpn-22 is also convincing. The paper needs no modifications. Publish as is.

Reviewer #3: In this report, the authors use a forward genetic screen in C. elegans to discover that loss of the tyrosine phosphatase ptpn-22 suppresses molting defects caused by partial loss of the kinases nekl-2 and nekl-3. They then use a proximity labeling approach to identify potential physical interacting partners of PTPN-22, including the putative GEF DNBP-1 and several other factors. They show that mutations in dnbp-1 also suppress nekl-2/3 defects, consistent with it acting as a partner for PTPN-22. They also show that several other hits exhibit genetic interactions with ptpn-22 in the context of other phenotypes.This study has several strengths:

The technical rigor of this study is high, especially the genetic analysis which makes use of multiple mutant alleles plus RNAi and genetic rescue experiments. The authors' conclusions are appropriately cautious, and they are careful to point out limitations in the approach and areas where the data do not fit neatly into a simple model.

Another strength is the use of TurboID proximity labeling, a technique that has only begun to be used in C. elegans in the last 2-3 years and for which only a handful of papers exist. This study provides a valuable guide and case study for this technique, including how to apply appropriate controls, how to filter the initial hit lists, and what kind of downstream analysis to perform to go from potential physical interactions to function.

The use of AlphaFold to validate potential physical interactions is interesting. While I would have expected to see an IP or two-hybrid experiment as follow-up from TurboID, the approach shown here may be the way of the future. AlphaFold may not be any less valid than biochemical assays that test binding under non-physiological conditions. In any case it sets an interesting example that may be further explored by others. The genetics showing that dnbp-1 also suppresses nekl-2/3 is a reassuring sign that the proposed physical interaction may be real.

Overall, the greatest value of this study is the demonstration of the power of combining genetics and proximity labeling to reinforce each other. This angle is what the authors chose to highlight in the abstract and throughout the text, a decision I agree with. The contribution of this study to our understanding of molting, Nekl kinases, or protein tyrosine phosphatase substrates is more limited, especially because 1) the authors have already identified other nekl-2/3 suppressors that may act more directly to oppose the loss of kinase activity and 2) it is unclear if any of the potential interactors are likely to be direct PTP substrates, given the lack of evidence such as altered phosphorylation state or genetic interaction with phosphomimic or non-phosphorylable point mutations.

I have several minor points that the authors should be able to address through revisions of the text or using reagents and assays already in hand.

1. Revisions in the text:

- Please explain earlier about serine/threonine (nekl-2/3) vs tyrosine (ptpn-22) phosphorylation. This did not come up until the Discussion, and it felt like a bait-and-switch as I had been thinking throughout the Results that DNBP-1 might be normally inactivated by NEKL-2/3 phosphorylation to promote molting, and dephosphorylated by PTPN-22.

- Along similar lines, please clarify the goal of the proximity labeling approach at the outset. From the Introduction, I had thought you were fishing for PTP substrates, but these seem likely to be more transient interactions. Were you trying to find phosphatase substrates? Regulators?

- At the authors' discretion it might help to include a "limitations" or "advantages and disadvantages" section somewhere to more formally summarize the use of TurboID in C. elegans. For example, the issue of detecting transient interactions or low-abundance interactors and possible work-arounds.

2. The authors seem to refer to the PTPN-22-TurboID fusion as "functional" referring to biotin ligase activity, but what about PTPN-22 activity? Does PTPN-22-TurboID fail to suppress nekl-2/3?

3. The authors use hyp7-specific expression of PTPN-22-TurboID to try to focus on interactors relevant to molting, but is this where PTPN-22 functions to counteract nekl-2/3? Does hyp7-specific expression of ptpn-22 rescue the ptpn-22; nekl-2/3 suppression? Points 2 & 3 might both be addressed using the existing Phyp7::PTPN-22::TurboID multicopy array.

4. The localization of PTPN-22 and DNBP-1 in Fig. 3F does not look that encouraging for a model in which they physically associate in this cell. Does co-localization become any more convincing during a molt? What developmental stage is the animal shown?

5. Please revise the statement that disrupting the putative PTPN-22-DNBP-1 interacting region did not lead to "strong suppression" (line 410). It looks like it does not suppress at all.

6. If possible, it would really help to include some additional supporting evidence that the interaction of PTPN-22 and DNBP-1 with each other or with CDC-42 is relevant to molting. Are hyperactive cdc-42 mutants suppressed by loss of dnbp-1 and ptpn-22? Or any other genetic, physical, or localization data would help.

7. While possibly beyond the scope of revisions, the use of the authors' phosphatase-dead point mutant as a substrate trap (see Flint et al. 1997 https://doi.org/10.1073/pnas.94.5.1680 ) for an additional round of TurboID would further strengthen this study. This might also be included as a discussion point if a "limitations" section is added.

**Have all data underlying the figures and results presented in the manuscript been provided?**

Reviewer #1: Yes

Reviewer #2: Yes

Reviewer #3: Yes

PLOS authors have the option to publish the peer review history of their article (what does this mean?). If published, this will include your full peer review and any attached files.

Reviewer #1: No

Reviewer #2: No

Reviewer #3: No

---

## [Editor Report · Decision Letter 1]

1 Aug 2024

Dear Dr Fay,

We are pleased to inform you that your manuscript entitled "A conserved protein tyrosine phosphatase, PTPN-22, functions in diverse developmental processes in C. elegans" has been editorially accepted for publication in PLOS Genetics. Congratulations!

Yours sincerely,

Anne C. Hart

Academic Editor

PLOS Genetics

Gregory P. Copenhaver

Section Editor

PLOS Genetics

Comments from the reviewers (if applicable):

**Data Deposition**

http://datadryad.org/submit?journalID=pgenetics&manu=PGENETICS-D-24-00285R1

**Press Queries**

---

## [Editor Report · Acceptance letter]

16 Aug 2024

PGENETICS-D-24-00285R1 

A conserved protein tyrosine phosphatase, PTPN-22, functions in diverse developmental processes in C. elegans 

Dear Dr Fay, 

We are pleased to inform you that your manuscript entitled "A conserved protein tyrosine phosphatase, PTPN-22, functions in diverse developmental processes in C. elegans" has been formally accepted for publication in PLOS Genetics! Your manuscript is now with our production department and you will be notified of the publication date in due course.

With kind regards,

Anita Estes

PLOS Genetics

On behalf of:
